# MET Exon 14 Skipping: A Case Study for the Detection of Genetic Variants in Cancer Driver Genes by Deep Learning

**DOI:** 10.3390/ijms22084217

**Published:** 2021-04-19

**Authors:** Vladimir Nosi, Alessandrì Luca, Melissa Milan, Maddalena Arigoni, Silvia Benvenuti, Davide Cacchiarelli, Marcella Cesana, Sara Riccardo, Lucio Di Filippo, Francesca Cordero, Marco Beccuti, Paolo M. Comoglio, Raffaele A. Calogero

**Affiliations:** 1Department of Molecular Biotechnology and Health Sciences, University of Torino, 10126 Torino, Italy; vladimir.nosi@unito.it (V.N.); alessandri.luca1991@gmail.com (A.L.); maddalena.arigoni@unito.it (M.A.); 2Candiolo Cancer Institute-FPO, IRCCS, 10060 Candiolo, Italy; melissa.milan@ircc.it (M.M.); silvia.benvenuti@ircc.it (S.B.); 3Telethon Institute of Genetics and Medicine (TIGEM), 80078 Pozzuoli, Italy; d.cacchiarelli@tigem.it (D.C.); m.cesana@tigem.it (M.C.); sara.riccardo@ngdx.eu (S.R.); lucio.difilippo@ngdx.eu (L.D.F.); 4Department of Computer Sciences, University of Torino, 10149 Torino, Italy; francesca.cordero@unito.it (F.C.); marco.beccuti@unito.it (M.B.); 5IFOM-FIRC Institute of Molecular Oncology, 20139 Milano, Italy; pcomoglio@gmail.com

**Keywords:** neural network, MET, exon skipping, genetic variants, deep learning

## Abstract

Background: Disruption of alternative splicing (AS) is frequently observed in cancer and might represent an important signature for tumor progression and therapy. Exon skipping (ES) represents one of the most frequent AS events, and in non-small cell lung cancer (NSCLC) MET exon 14 skipping was shown to be targetable. Methods: We constructed neural networks (NN/CNN) specifically designed to detect MET exon 14 skipping events using RNAseq data. Furthermore, for discovery purposes we also developed a sparsely connected autoencoder to identify uncharacterized MET isoforms. Results: The neural networks had a Met exon 14 skipping detection rate greater than 94% when tested on a manually curated set of 690 TCGA bronchus and lung samples. When globally applied to 2605 TCGA samples, we observed that the majority of false positives was characterized by a blurry coverage of exon 14, but interestingly they share a common coverage peak in the second intron and we speculate that this event could be the transcription signature of a LINE1 (Long Interspersed Nuclear Element 1)-MET (Mesenchymal Epithelial Transition receptor tyrosine kinase) fusion. Conclusions: Taken together, our results indicate that neural networks can be an effective tool to provide a quick classification of pathological transcription events, and sparsely connected autoencoders could represent the basis for the development of an effective discovery tool.

## 1. Introduction

It is known that in eukaryotes, alternative splicing plays an important role in defining the protein diversity and enhancing the complexity of gene expression regulation [1]. In humans, the majority of multi-exon genes is affected by alternative splicing, which generates proteins with different functions in distinct cellular processes [2]. Alteration of alternative splicing (AS) is associated with human diseases [3] and exon skipping (ES) is one of the most observed events [4]. The analyses and studies of alternative splicing advance our understanding of mRNA complexity and its regulation, providing valuable insights to grasp disease etiology, and assisting the development of therapeutic interventions for splicing-related diseases [5]. ExonSkipDB (https://ccsm.uth.edu/ExonSkipDB/, accessed on 1 October 2020) [6] has been recently developed, which is a database collecting ES events affecting disease associated genes. Within the 8266 ExonSkipDB genes, annotated as genes loosing functional features due to in-frame ES events in TCGA (https://portal.gdc.cancer.gov/, accessed on 1 March 2020), 449 are part of the 710 COSMIC census genes [7]. 25 of them (ALK (Anaplastic Lymphoma Kinase), APC (Adenomatous Polyposis Coli), BAP1 (BRCA1 Associated Protein 1), BRCA1 (Breast And Ovarian Cancer Susceptibility Protein 1), BRCA2 (Breast And Ovarian Cancer Susceptibility Protein 2), BRIP1 (BRCA1-Interacting Protein C-Terminal Helicase 1), BTK (Bruton Tyrosine Kinase), CDH1 (Cadherin 1), CHEK2 (Checkpoint Kinase 2), ERBB2 (Erb-B2 Receptor Tyrosine Kinase), ETV6 (ETS Variant Transcription Factor 6), EXT1 (Exostosin Glycosyltransferase 1), EZH2 (Enhancer Of Zeste 2 Polycomb Repressive Complex 2 Subunit), GLI1 (Glioma-Associated Oncogene Homolog 1), JAK2 (Janus Kinase 2), MDM4 (MDM4 Regulator Of P53), MET (Mesenchymal Epithelial Transition receptor tyrosine kinase), MLH1 (MutL Homolog 1), MUTYH (MutY DNA Glycosylase), NF2 (Neurofibromin 2), NOTCH2 (Neurogenic Locus Notch Homolog Protein 2), PIK3R1 (Phosphoinositide-3-Kinase Regulatory Subunit 1), PTCH1 (Patched 1), SUFU (Suppressor Of Fused Homolog), TP53 (Tumor Protein P53)) have at least an exon skipping event associated with cancer phenotype reported in at least a published paper (Appendix A). 

Notably, MET exon 14 skipping is the only ES event encompassing a massive number of citations (119 from 2015 to 2021 reported in the PUBMED repository). MET exon 14 skipping is a splicing aberration that results in deletion of the MET juxtamembrane domain, which contains negative regulatory sites of the MET receptor. Thus, exon 14 deletion results in impaired receptor ubiquitin-mediated receptor degradation, decreased turnover and increased downstream signaling [8,9]. MET exon 14 skipping was described in lung adenocarcinoma (3%), other lung neoplasms than adenocarcinomas (2.3%), brain glioma (0.4%), and tumors of unknown primary origin (0.4%) [8]. Furthermore, Champagnac and coworkers [10] observed that genomic alterations affecting MET exon 14 are present in 2.6% of non-small cell lung cancer (NSCLC) patients. MET exon 14 skipping can lead to acquisition of transforming ability and was identified as a potential therapeutic target for NSCLC [11,12]. Many different mutations at the DNA level can cause the aberrant splicing of exon 14, and the only search at the genomic level for MET exon 14 skipping does not guarantee that the mutated MET transcript is actively expressed. Furthermore, given the relatively small deletion, it remains a question as to whether antibodies can be developed with enough specificity against this splice variant [13]. RNA sequencing is today a straightforward approach thanks to the possibility to perform targeted RNAseq in paraffin embedded samples [14]. However, to efficiently detect MET exon 14 skipping, an effective computing detection algorithm for this specific ES event is also required. Inspecting the available literature on MET exon 14 skipping (https://pubmed.ncbi.nlm.nih.gov/, accessed on 01 March 2021) we observed that the identification of this skipping event is mainly done using DNA-based amplicon-mediated target enrichment [15] or RNA-based next-generation sequencing target enrichment [16], where the RNA based method provides a higher detection rate of exon 14 skipping [16]. We never found any article using deep learning or machine learning methods for the detection of MET exon 14 skipping, which are of particular interest to screen large cohorts of specimens. Notably, we found two articles [17,18] predicting exon skipping events using RNAseq data. In Zhang’s paper [17] a convolutional neural network (CNN) is used to classify splice junctions derived from primary RNA-seq data. Instead, in Du’s paper [18] a Rotation Forest algorithm is used to predict ES events integrating RNA-Seq data and genome sequence information. Moreover, CNN was implemented in SpliceRover [19] a generalist tool for splice site prediction. Similarly, in [20] a general tool—namely, SpliceAI—was proposed to predict splicing from a pre-mRNA sequence using CNN.

To the best of our knowledge, we did not find any tool designed specifically to detect an exon skipping event as MET exon 14 skipping. In this manuscript, we investigated different neural network architectures to provide sensitive and rapid detection of MET exon 14 skipping events using RNAseq data. Standard Neural Network (NN), Convolutional Neural Network (CNN), and a Sparsely Connected Autoencoder (SCA) were thus compared in detail on different datasets. With respect to models predicting ES events, which are designed using nucleotide sequence information for their prediction, our models are designed to handle expression data in the form of kmer counts or coverage.

## 2. Results

### 2.1. Neural Network for the Detection of MET Exon 14 Skipping (METΔ14).

To detect MET exon 14 skipping events, an NN made of six layers was built; Figure 1A.

As a training set for the NN, we used data from amplified WT MET and exon 14 skipping MET (Table 1).

Specifically, we split the MET reads in random non-overlapping subgroups of 1000 reads. Although at 1000 reads, coverage of the detection of METΔ14 becomes a bit blurry—Figure 2D; this threshold allows for a generation of large number of MET (1447) and METΔ14 (846) to not overlap subsamples, and a high numerosity of training data is an important element for efficient learning of the NN.

Each of the above-mentioned subgroups was converted in 31 and 16 k-mers. MET expression was represented by the amount of each k-mers spanning over MET exons, and these data were used to train the NN. We observed that the learning curve at 16 k-mers was slightly better than the one at 31 k-mers (not shown), and thus we ran the following analyses using the 16 k-mers representation of MET. As training sets, we also used k-mer count frequency [21] for full MET locus, k-mer count frequency for MET exons 13 ÷ 15 and coverage frequency for MET exons 13 ÷ 15.

As test sets, we used subsets of WT and exon 14 skipping MET from cell lines characterized by a physiological MET expression. NN performance was investigated using as test set: (i) subsets made of random not overlapping subgroups of 500, 1000 and 5000 reads, converted in 16 k-mer counts, (ii) k-mer count frequency on full MET locus, (iii) k-mer count frequency on MET exons 13 ÷ 15 and (iv) coverage frequency for MET exons 13 ÷ 15.

The detection efficiency of METΔ14 using 16 k-mers frequency counts showed best performances at 500 and 1000 reads coverage, Figure 3A,B, as instead at 5000 reads coverage of all the different test sets performed in the same way, Figure 3C.

#### Neural Network Validation and Discovery on TCGA Samples

To validate the METΔ14 discovery potential of the above-described NN, we used a set of 690 RNAseq samples from the TCGA bronchus and lung dataset. The 690 samples were manually inspected using the Broad’s integrative genomics viewer (IGV) [22] and we detected 17 exon 14 skipping events (2.4%), which is in line with the frequency of the exon 14 skipping events observed in published literature [8,10]. We tested on this tumor set the NN trained with k-mer counts frequency, which predicted 4 samples out of 17 as METΔ14, but only one was a real exon skipping event (sensitivity 5.88%, specificity 99.5%). The NN trained with exons 13 ÷ 15 MET k-mer counts frequency improved the detection of METΔ14 events, 9 out of 17 (sensitivity 52.9%), but this prediction included a massive increase of false positives, 129 samples, (specificity 81.3%). The best results were obtained using the NN trained using only the coverage frequency for MET exons 13 ÷ 15, which predicted 18 skipping events, including all 17 true skipping events (sensitivity 100%) and one false positive (specificity 99.8%).

Using the NN trained with the coverage frequency for MET exons 13 ÷ 15, we extended the METΔ14 discovery to 2605 TCGA tumor tissues; Table 2.

We could detect only one METΔ14 in 280 bladder samples. Then, we detected few false METΔ14 in cervix, corpus uteri, heart/mediastinum/pleura, kidney and skin samples, Table 2. The six transcripts detected in cervix, Table 2, were erroneously detected as METΔ14, because they have a blurry coverage on exons 13 ÷ 15, Figure 4C. However, when the full MET locus is observed, it is clear that these METΔ14 false positives are a completely different type of transcript. A shared characteristic of these transcripts is the high accumulation of reads in the second intron (approx. chr7:116,715,690–116,717,329), Figure 4A, in the 6th exon, Figure 4B, and in the last non-coding MET exon, Figure 4D.

The above observation also applies to the other false METΔ14 detected in corpus uteri, heart/mediastinum/pleura, kidney and skin samples; supplementary Appendix A. 

A possible explanation could be that we are observing the transcriptional effect of a LINE1-MET fusion, which was firstly described a few years ago in triple negative breast cancers [23]. We further investigated this point searching for LINE1 alignment, in the subset of MET reads, where only one of the two pair-end reads maps on MET. Indeed, in 10 out of 15 samples, detected as characterized by a transcription peak in MET second intron, we detected LINE1 mapping reads, Appendix A. From the samples shown in Appendix A, we extracted the paired reads associated with MET reads, i.e., only one read of the pair is mapping in MET locus. We blasted [24] these reads on a LINE1 sequence (chr1:62194249–62212928, hg38) and indeed, some of these reads map to LINE1 sequence; supplementary Appendix A. On the basis of the MET read position we could identify the putative fusion point with MET, which is mainly located in MET intronic regions and in the last non-coding exon. Unfortunately, we cannot pair the TCGA RNAseq samples to genomics data to further validate the presence of a LINE1 insertion on the basis of genome sequencing data.

### 2.2. Convolutional Neural Network (CNN) for the Detection of METΔ14

To detect MET exon 14 skipping events, we constructed a CNN made by a 1D convolutional layer, 1D Max pooling layer, a flat fully connected dense layer with 50 nodes and an output layer with one node; Figure 1C. The CNN was challenged with the same training and test set used for the flat neural network. In this implementation, the convolutional layer included 10 kernels, for more information see Material and Method section. In Figure 5, the METΔ14 detection ability of CNN on the basis of different representation of the MET expression data are reported.

The results are organized (Figure 5) on the basis of the type of input data, i.e., whole MET exons kmer counts (Figure 5A), MET exons 13 ÷ 15 kmer count frequency (Figure 5B) and MET exons 13–15 coverage frequency (Figure 5C). The best ratio between true positive and false positive is shown for all kernels using test samples characterized by 5000 reads coverage. As also seen for NN (Figure 3), the specificity progressively decreases when the coverage is reduced.

#### Convolutional Neural Network Validation on Bronchus and Lung Samples

We validated the CNN model using the kernel 100, which is one of the best performing kernels independently by the coverage of the test set (Figure 5). The validation was done on the 690 TCGA bronchus and lung sample manually inspected for the presence of METΔ14. We tested on this tumor set the CNN trained with k-mer counts, which predicted 10 samples as METΔ14, but only one was a real exon skipping events (sensitivity 5.88%, specificity 97.6%, supplementary Appendix A). The best results were obtained using the CNN trained with exons 13 ÷ 15 MET k-mer counts frequency. All of the 16 samples predicted as METΔ14 belong to the 17 true METΔ14 present in the data set (sensitivity 94.11%, specificity 100%, supplementary Appendix A). Finally, the CNN trained using only the coverage frequency for MET exons 13 ÷ 15, predicted 8 skipping events and all of them belong to the true skipping events (sensitivity 47.05%, specificity 100%, supplementary Appendix A). Since we observed that NN was detecting some false positives in cervix tumor tissues (Figure 4), we evaluated if CNN was more specific than NN. CNN trained with exons 13 ÷ 15 MET k-mer counts frequency detects the same false positives detected by NN, Figure 4.

### 2.3. Sparsely Connected Autoencoders (SCA) to Detect MET Non-Canonical Isoforms

Our group have recently published a paper on the use of SCA for the identification of hidden functional regulatory elements in single cell RNAseq data [25]. We tested this type of autoencoder to see if we could grasp non-canonical isoforms from the analysis of the TCGA samples used in the previous paragraph. The SCA was designed to take as input k-mer count frequency or coverage frequency of MET exons. The SCA hidden layer, i.e., latent space, is representing MET exons. Input nodes are only connected to the exon nodes they are associated (Figure 1B). We trained the SCA with the 2605 TCGA samples and clustered the latent space data using gridFLOW [26]. To estimate the stability of clusters generated using the SCA latent space, we compared thousands of pairs of clusters generated by SCA latent space clustering, as previously described by us [25]. The rationale of this approach is that, if a cluster’s organization is conserved, it should be depicted by the multiple comparisons of randomly paired latent space cluster representations [25]. The best results were obtained using normalized [27] MET coverage frequency data (Figure 5A). Unfortunately, the stability of the clusters was very poor (Figure 6A). However, an inspection of a random subsets of samples associated with cluster 2 (Figure 6B) suggests that at least cluster 2 seems to be made mainly of transcripts recalling the organization of MET-LINE1 fusion, which we have described in previous paragraphs.

## 3. Discussion

We used MET exon 14 skipping as a case study for the detection of genetic variants in cancer driver genes through deep learning. In recent years, a lot of evidence has indicated that MET inhibitors have a good anti-tumor effect in patients with MET exon 14 skipping mutation, suggesting that MET exon 14 skipping may be a new target for NSCLC patients [28]. Thus, the availability of effective tools for the detection of MET exon 14 skipping are needed for a fast identification of patients suitable for MET targeted therapy. 

It is notable that, digging into the published literature, all the found exon skipping tools use nucleotide sequence analysis to infer skipping events, and they are only able to predict skipping events in a generalist way. Since we could not find any tool providing the detection of a unique skipping event in a gene over a large cohort of specimens, we designed specific neural networks for the identification of MET exon 14 skipping, using transcript expression information. 

We designed a conventional neural network (NN) made of four fully connected hidden layers and a convolutional neural network (CNN) made of one 1D convolutional layer, one 1D max pooling layer and a fully connected dense layer. Although we performed an automated optimization of the hyperparameters, the prediction efficacy of our CNN and NN comes from the special attention we put on defining the optimal representation of the data for each architecture, i.e kmer counts for CNN and coverage from NN.

The NN and the CNN training was done using the RNAseq data of a lung cancer cell line expressing amplified form of the wild type MET (WT, EBC-1), and a gastric cancer cell line expressing exon 14 skipped MET (HS746T). HS746T cell line was selected because, to the best of our knowledge, it is the only cell line displaying amplification of MET exon 14 skipping isoform. MET gene amplification has been observed in about 2–5% of gastroesophageal cancers and represents an oncogenic driver and therapeutic target [29,30]. MET exon 14 skipping was initially described in NSCLCs (caused by a mutation in the splice donor site in intron 14 and afterwards reported in a variety of tumors, including gastrointestinal cancers, suggesting it as a potential mechanism leading to MET activation [31]. Therefore, HS746T, together with EBC-1, were invaluable instruments to provide a large amount of data for the NN/CNN training. Validation was done instead using RNAseq data from lung cancer cell lines expressing at physiological level MET (A549 expressing WT MET and NCI-H596 expressing exon 14 skipped MET).

Since we could not compare our models with respect to pre-existing methods for MET exon 14 skipping, we manually curated a set of TCGA data to provide an objective evaluation of the performance of our tool. Specifically, we manually curated a cohort of WT and exon 14 skipped samples made of the 690 RNAseq samples belonging to the TCGA (https://www.cancer.gov/tcga, accessed on 1 March 2020) bronchus and lung collection (1310 samples) showing a MET coverage of at least 5000 reads. Given the manual curation of this dataset, i.e., each single sample was inspected on IGV browser for the presence of MET exon 14 skipping, it represents a robust instrument to quantify the predictive performance of our neural network models.

Skewed datasets are not uncommon and the MET exon 14 skipping detection is a typical example. Although skewed datasets are tough to handle, our models, i.e., CNN and NN, seem to handle this issue efficiently, since sensitivity greater than 94% and specificity greater than 99% are reached on an extremely skewed data set such as TCGA bronchus and lung 690 samples with only 17 MET exon skipping events (2.46%). Notably, the high sensitivity is obtained by CNN with a training based on kmer counts spanning among MET exon 13 and exon 15. Instead, in the case of the NN the optimal sensitivity was obtained with a training based on coverage data encompassing the region among MET exon 13 and exon 15.

Our analysis, using both CNN and NN, on 2605 TCGA tumors (13 primary sites, Table 2) highlights that MET exon 14 skipping is a peculiar event of lung specimens. Then, mainly in uterine cancers, we detected a set of MET exon 14 skipping false positives, sharing a common feature: an unexpected peak of coverage in the MET intron 2. This observation brought us to speculate that we were observing a transcriptional signature for a LINE1-MET fusion event [23]. This hypothesis has been supported by the identification of MET paired-end reads, having one read mapping on MET and the other on LINE1 sequence. Notably, transcription of the LINE1-MET fusion was observed in advanced stages of cancer [23,32], but very little is still known about the effect of the LINE1-MET chimera in cancer. 

At the present time, we cannot manage to eliminate LINE1-MET false positives, mainly because we do not have enough data to train a model to detect LINE1-MET fusion, to be implemented in parallel with the MET exon 14 skipping models. However, we are generating large RNAseq data from MCF7, a breast cancer cell line harboring LINE1-MET fusion [23], to build a specific CNN to be integrated with our MET exon 14 skipping models, to improve their specificity.

Having identified more than one artefactual event in MET, we investigated the possibility to discover those anomalous events by the integration of a particular type of deep learning tool, sparsely connected autoencoders [25], with clustering techniques used in multicolor cytometry. Although the actual implementation of the SCA tool could be further improved in terms of its precision and sensitivity, currently we were able to detect from TCGA specimens a set of tumors sharing the putative LINE1-MET fusion.

Taken together, our results indicate that neural networks can be an effective tool to provide a quick classification of pathological transcription events. However, from the discovery point of view there is still some work to be done to obtain an effective discovery tool using sparsely connected autoencoders.

## 4. Materials and Methods

### 4.1. Cell Lines

A549 (lung adenocarcinoma); NCI-H596 (lung adenocarcinoma); Hs746T (gastric adenocarcinoma) cell lines were purchased from ATCC (Rockville, MD, USA); EBC-1 (non-small cell lung cancer) were acquired from HSRRB cell bank (Osaka, Japan). All cells were kept in culture for less than 4 weeks and used between passage 2 and 10. Cells were grown in recommended media (Sigma Aldrich, St. Louis, MO, USA) supplemented with 50 units/mL penicillin (Sigma Aldrich, St. Louis, MO, USA), 50 mg/mL streptomycin (Sigma Aldrich, St. Louis, MO, USA), 2 mM glutamine (Sigma Aldrich, St. Louis, MO, USA) and 10% Foetal Bovine Serum (Lonza Sales Ltd., Basel, Switzerland) as indicated. Cells were maintained at 37 °C in a 5% CO_2_ atmosphere.

### 4.2. Generating the Data for the Neural Network Training and Test Set

We generated RNAseq data from EBC-1 [33], a non-small cell lung cancer (NSCLC) cell line, harboring MET amplification and from Hs746T, a gastric cancer cell line, harboring amplified MET exon 14 skipped isoform (METΔ14) [34]. Furthermore, we have performed RNAseq on human lung adenocarcinoma cell line A549, expressing c-Met [35] and on NCI-H596, derived from an NSCLC, expressing exon 14 skipped MET [36], Table 1. Both cell lines express physiological levels of MET.

Total RNA was extracted from cell lines using Trizol reagent (Invitrogen, Carlsbad, CA, USA), following the manufacturer indication. Total RNA was quantified using the Qubit 2.0 fluorimetric Assay (Thermo Fisher Scientific, Waltham, MA, USA) and sample integrity, based on the RIN (RNA integrity number), was assessed using an RNA ScreenTape assay on TapeStation 4200 (Agilent Technologies, Santa Clara, CA, USA).

Libraries were prepared from 400 ng of total RNA using the RNAseq (total RNA Full length) sequencing service (Next Generation Diagnostics srl) which included rRNA-Globin depletion, library preparation, quality assessment and sequencing on a NovaSeq 6000 sequencing system using a paired-end, 300 cycle strategy (2 × 150) (Illumina Inc., San Diego, CA, USA). Read were trimmed to remove adapters sequences using skewer (https://github.com/relipmoc/skewer accessed on 1 March 2021) Read were mapped using STAR on ENSEMBL HG38 human genome assembly.

#### 4.2.1. 16/31. k-mer Training Set

The training set for the neural networks (NN/CNN) was generated using the cell lines with MET amplification: EBC-1 and Hs746T. EBC-1 and Hs746T reads were organized in subgroups of 1000 reads, randomly selected and not overlapping. This approach generated a large set of samples for the training the NN, i.e., 1447 subsets for EBC-1 and 846 for Hs746T. Subsampled reads were associated to MET exons (supplementary Appendix A) and converted in 16/31 k-mers using BFcounter [37].

#### 4.2.2. 16/31. k-mer Test Set

The test set for the neural networks (NN/CNN) was generated using the cell lines with physiological MET expression: A549 and NCI-H596. A549 and NCI-H596 reads were organized in subgroups of 500, 1000 and 5000 reads, randomly selected and not overlapping. Subsampled reads were associated with MET exons (supplementary Appendix A) and converted in 16/31 k-mers using BFcounter [37].

#### 4.2.3. Coverage Training and Test Set

The training and test set for the neural networks (NN/CNN) was generated using the RNAseq data used for the 16/31 k-mers training and test sets. For the training, reads were organized in subgroups of 1000 reads, randomly selected and not overlapping, for the test set, reads were organized in subgroups of 500, 1000 and 5000 reads. Subsampled reads were used to calculate coverage associated with MET exons 13, 14 and 15.

### 4.3. TCGA RNAseq Datasets

We registered a TCGA project for the study of MET exon 14 skipping events, to obtain access to TCGA raw sequencing data, i.e., RNAseq BAM files. Since the size of the TCGA transcription data exceeds 200 TB, we progressively downloaded the BAM files on the basis of the cancer tissue locus. Then, from each BAM file, we extracted the reads encompassing MET locus (chr7:116672196–116798377, hg38 human genome assembly). We kept only samples where the MET locus was covered by at least 5000 reads. To define 5000 reads as the minimal coverage for MET, we inspected the expected coverage for exons 13, 14 and 15 in A549 (WT MET cell line), in NCI-H596 (METΔ14 cell line) and in random subsets of 5000, 1000, 500 and 250 reads from NCI-H596, Figure 2. We observed that the detection of exon 14 skipping become blurry below 5000 reads coverage. Together with the MET linked reads we also extracted the MET paired reads, where only one of the two reads maps on MET locus.

### 4.4. Model Coding and Hyperparameter Selection for NN

We constructed a NN made of 6 layers. The input layer has variable size depending on the type of input (k-mers or coverage). 1st and 2nd hidden layers are made of 256 nodes, 3rd and 4th are made of 128 nodes, all using RELU (rectified linear unit) as activation function and 0.1 as dropout rate. The output layer is made by 1 node, associated with a sigmoid activation function. We implemented the models in python (version 3.7) using TensorFlow package (version 2.0.0), Keras (version 2.3.1), pandas (version 0.25.3), numpy (version 1.17.4), matplotlib (version 3.1.2), sklearn (version 0.22), scipy (version 1.3.3). Optimization was done using Adam (Adaptive moment estimation), with the following parameters lr = 0.01, beta_1 = 0.9, beta_2 = 0.999, epsilon = 1e−08, decay = 0.0, loss = ‘mean_squared_error’. Hyperparameter optimization was done using Talos (https://github.com/autonomio/talos, accessed on 01 January 2021), which is an automated tool to define the optimal combination of the hyperparameters. Specifically, Talos takes as input the hyperparameter space to be investigated. Then, Talos performs all possible combinations and selects the optimal configuration of the hyperparameters.

The trained NN is implemented in a docker container together with all tools needed to extract MET reads from fastq data. The NN can be used for the discovery of METΔ14 using conventional RNAseq or MET targeted RNAseq. The tool can be requested for the corresponding author. It is provided free of charge to Accademia and non-profit organizations for research use only.

### 4.5. Model Coding and Hyperparameter Selection for CNN

We constructed a CNN made of one Convolutional 1D layer, characterized by 64 filters and the following kernel sizes: 2, 5, 7, 10, 15, 50, 75, 100, 150, 200. One MaxPooling 1D layer with pool size of 2, for dimensionality reduction. One dense layer with activation RELU and 50 nodes and one dense layer with activation sigmoid and 1 node. We implemented the models in python (version 3.7) using TensorFlow package (version 2.0.0), Keras (version 2.3.1), pandas (version 0.25.3), numpy (version 1.17.4), matplotlib (version 3.1.2), sklearn (version 0.22), scipy (version 1.3.3). Optimization was done using Adam (Adaptive moment estimation) using the following parameters lr = 0.01, beta_1 = 0.9, beta_2 = 0.999, epsilon = 1e−08, decay = 0.0, loss = ‘mean_squared_error’. Hyperparameter optimization was done using Talos as done for NN.

### 4.6. Model Coding and Hyperparameter Selection for Sparsely Connected Autoencoders (SCA)

Autoencoders learning is based on an encoder function that projects input data onto a lower dimensional space. Then, autodecoder function recovers the input data from the low-dimensional projections minimizing the reconstruction. We implemented the models in python (version 3.7) using TensorFlow package (version 2.0.0), Keras (version 2.3.1), pandas (version 0.25.3), numpy (version 1.17.4), matplotlib (version 3.1.2), sklearn (version 0.22), scipy (version 1.3.3). Optimization was done using Adam (Adaptive moment estimation) with the following parameters lr = 0.01, beta_1 = 0.9, beta_2 = 0.999, epsilon = 1e−08, decay = 0.0, loss = ‘mean_squared_error’. RELU (rectified linear unit) was used as activation function for the dense layer.

## 5. Conclusions

Taken together, our results indicate that neural networks can be an effective tool to provide a quick classification of pathological transcription events, and sparsely connected autoencoders could represent the basis for the development of an effective discovery tool in this field.

## Figures and Tables

**Figure 1 ijms-22-04217-f001:**
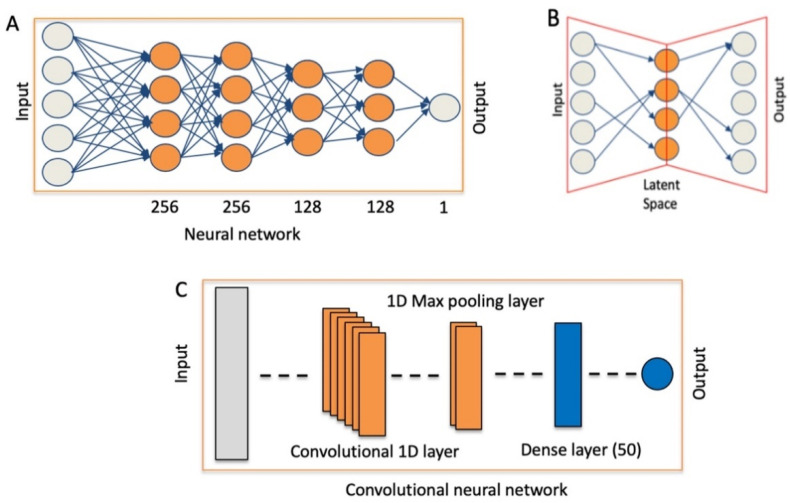
Neural networks used for the analysis of MET variants. (**A**) NN for the detection of exon 14 skipping events. (**B**) Sparsely-connected autoencoders for the detection of MET transcription variants. (**C**) Convolutional neural network (CNN).

**Figure 2 ijms-22-04217-f002:**
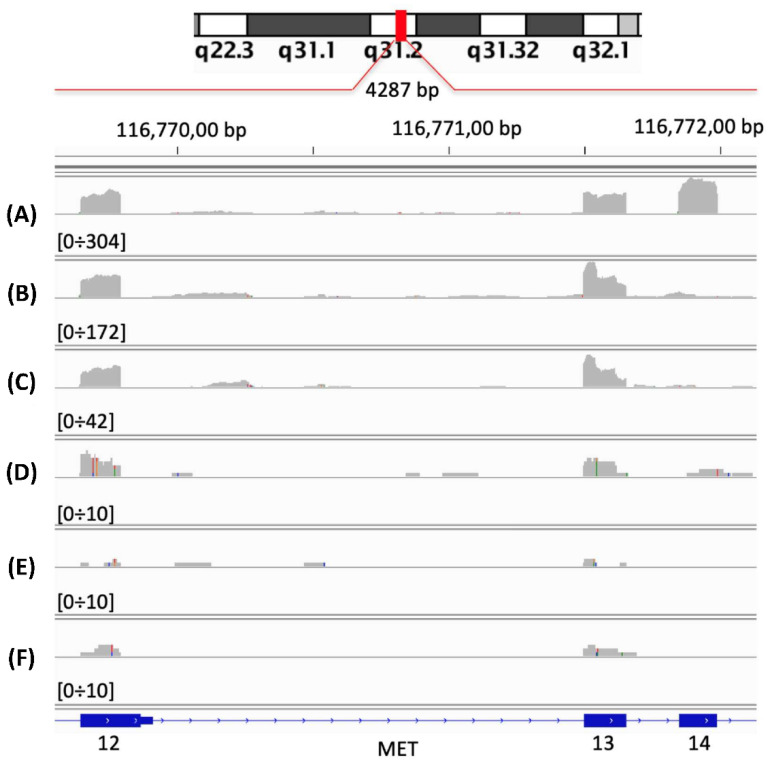
Expected coverage for exons 13, 14 and 15. (**A**) WT MET from A549 RNAseq sample (33 million reads), 27,152 reads mapping on MET locus, (**B**) METΔ14 from NCI-H596 RNAseq sample (27 million reads), 24,850 reads mapping on MET locus, (**C**) 5000 reads randomly selected from (**B**), (**D**) 1000 reads randomly selected from (**B**), (**E**) 500 reads randomly selected from (**B**), (**F**) 250 reads randomly selected from (**B**).

**Figure 3 ijms-22-04217-f003:**
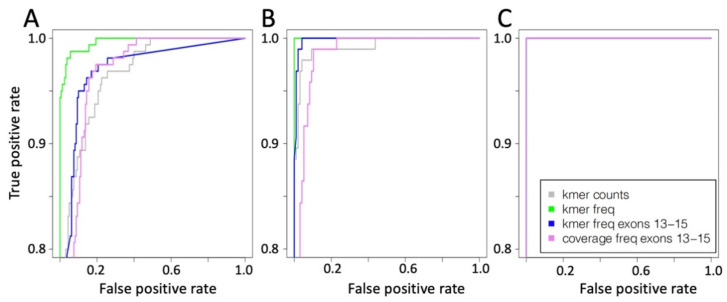
ROC curve for NN prediction. (**A**) Training based on 1000 reads coverage for MET locus and test with a coverage of 500 reads. (**B**) Training based on 1000 reads coverage for MET locus and test with a coverage of 1000 reads. (**C**) Training based on 1000 reads coverage for MET locus and test with a coverage of 5000 reads. Grey line training and test using k-mer counts, green line training and test using k-mer counts frequency, blue line training and test using k-mer counts frequency only for exons 13 ÷ 15, violet line training and test using coverage counts frequency only for exons 13 ÷ 15.

**Figure 4 ijms-22-04217-f004:**
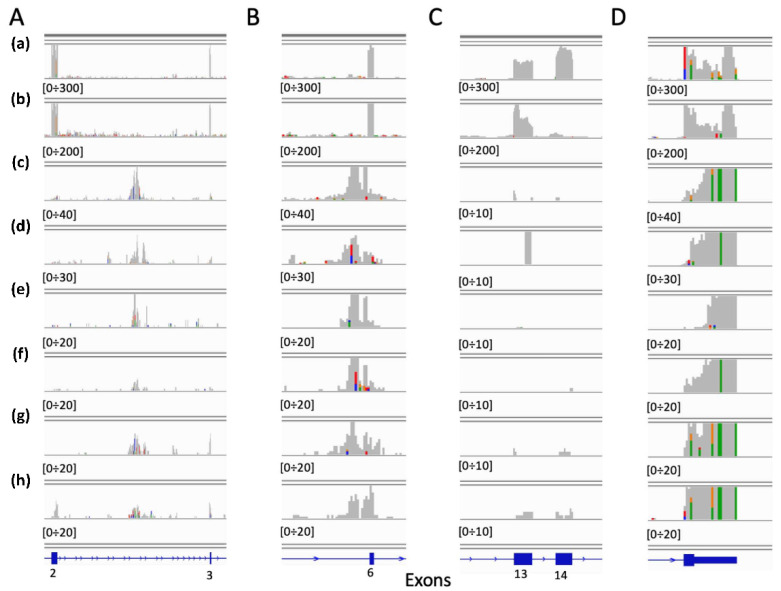
METΔ14 false positive detected in cervix. (a) WT MET from A549 RNAseq sample (33 million reads), 27,152 reads mapping on MET locus, (b) METΔ14 from NCI-H596 RNAseq sample (27 million reads), 24,850 reads mapping on MET locus, (c–h) False METΔ14. (**A**) Zoom in the 2–3 exons region. (**B**) Zoom in 6th exon region. (**C**) Zoom in 13–15t exons region. (**D**) zoom in last exon region.

**Figure 5 ijms-22-04217-f005:**
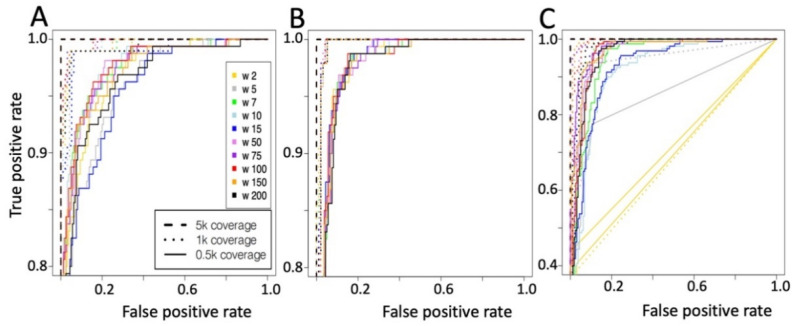
ROC curve for CNN prediction. (**A**) 16 kmer counts frequency for whole MET exons. (**B**) 16 kmer counts frequency for MET 13 ÷ 15 exons. (**C**) Coverage frequency for MET 13 ÷ 15 exons.

**Figure 6 ijms-22-04217-f006:**
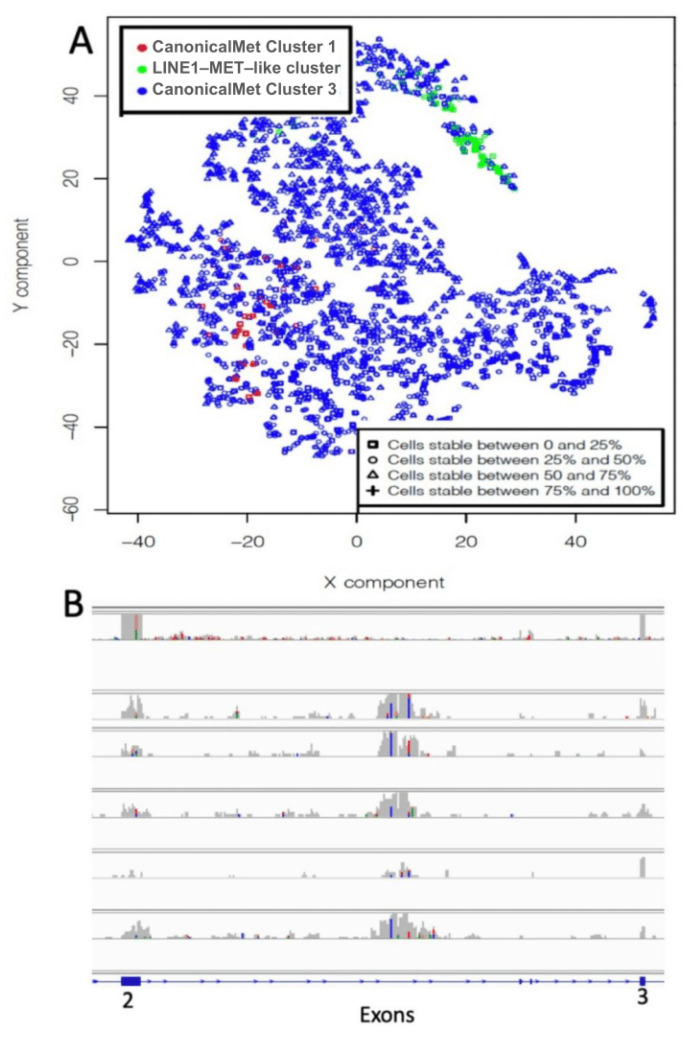
Autoencoder on saver normalized data. (**A**) Clustering results of the latent space trained with 2605 TCGA samples. (**B**) A limited number of samples (green group in A) is characterized the presence of a transcription patter, i.e., the coverage peak in intron 2, which resembles the presence of a LINE1-MET fusion. In B, it is shown a set of samples randomly picked from cluster 2.

**Table 1 ijms-22-04217-t001:** MET cell line RNAseq data.

Cell Line	Status	RNAseq (Million Reads)	MET (Thousand Reads)
EBC-1	Amplified MET	113	1447
Hs746T	Amplified METΔ14	95	846
A549	MET	115	109
NCI-H596	METΔ14	118	114

**Table 2 ijms-22-04217-t002:** TCGA samples inspected for the presence of METΔ14.

TCGA Tissue	# Inspected Tissue	# Detected METΔ14	# Detected False METΔ14
Adrenal gland	10	0	0
Bladder	280	1	0
Brain	28	0	0
Breast	162	0	0
Bronchus and lung	690	17	1
Cervix (uterus)	236	0	6
Corpus uteri	109	0	4
Esophagus	165	0	0
Hearth/mediastinum/pleura	78	0	1
Kidney	435	0	3
Pancreas	89	0	0
Skin	288	0	1
Soft tissues	35	0	0

## Data Availability

Examples and instructions for the use of the NN/CNN tools are available at https://github.com/kendomaniac/metObservatory (accessed on 13 April 2021). RNAseq used for training and test data are available at https://github.com/kendomaniac/metObservatory (accessed on 13 April 2021).

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
