# Peer review of "MET Exon 14 Skipping: A Case Study for the Detection of Genetic Variants in Cancer Driver Genes by Deep Learning"

_ijms, 2021, doi:10.3390/ijms22084217_

Round 1

Reviewer 1 Report

In this study, Nosi et al. proposed a deep learning approach to detect genetic variants in cancer driver genes. A case study of MET exon 14 skipping has been tested and a promising performance has been reached. There are some major issues that need to be addressed:

1. In the "Introduction", the authors should add more literature reviews on the studies related to their study on genetic variant prediction using deep learning.

2. A main point/contribution of this paper is the use of deep learning. However, the authors did not mention it a lot in the methodology as well as in the results. More analyses on deep learning architecture should be performed.

3. The authors should compare the predictive performance of their models with previous studies on the same problem/dataset. Also, some baseline comparisons should be conducted to convince that the authors' choice is the optimal one.

4. How did the author deal with hyperparameter optimization of the models? It should be described clearly.

5. How did the authors deal with imbalance problem? Sometimes I have seen that the authors reached a very imbalance result (supplementary table S2).

6. The tool cannot be accessed. Please re-check the link.

7. Deep learning, or DNN, has been used in previous studies i.e., PMID: 33260643 and PMID: 31750297. Therefore, the authors are suggested to refer to more works in this description.

8. Quality of figures should be improved significantly.

9. More validation data should be used.

Author Response

We would like to thank the reviewers for their valuable comments and useful suggestions that helped us to substantially improve the quality of this paper

Reviewer 1

Comments and Suggestions for Authors

In this study, Nosi et al. proposed a deep learning approach to detect genetic variants in cancer driver genes. A case study of MET exon 14 skipping has been tested and a promising performance has been reached. There are some major issues that need to be addressed:

Q1: In the "Introduction", the authors should add more literature reviews on the studies related to their study on genetic variant prediction using deep learning.

Answer: We extended the introduction:

“... Notably, MET exon 14 skipping is the only ES event encompassing a massive number of citations (119 from 2015 to 2021 reported in the PUBMED repository).  MET exon 14 skipping is a splicing aberration that results in deletion of MET juxtamembrane domain which contains negative regulatory sites of the MET receptor. Thus, exon 14 deletion results in impaired receptor ubiquitin-mediated receptor degradation, decreased turnover and increased downstream signaling [8, 9]. MET exon 14 skipping was described in lung adenocarcinoma (3%), other lung neoplasms than adenocarcinomas (2,3%), brain glioma (0.4%), and tumors of unknown primary origin (0.4%) [8]. Furthermore, Champagnac and coworkers [10] observed that genomic alterations affecting MET exon 14 are present in 2.6% of non-small cell lung cancer (NSCLC) patients. MET exon 14 skipping can lead to acquisition of transforming ability and was identified as a potential therapeutic target for NSCLC [11, 12]. Many different mutations at DNA level can cause the aberrant splicing of exon 14, and the only search at genomic level for MET exon 14 skipping does not guarantee that the mutated MET transcript is actively expressed. Furthermore, given the relatively small deletion, it remains a question whether antibodies can be developed with enough specificity against this splice variant [13]. RNA sequencing is today a straightforward approach thanks to the possibility to perform targeted RNAseq in paraffin embedded samples [14]. However, to efficiently detect MET exon 14 skipping an effective computing detection algorithm for this specific ES event is also required. Inspecting the available literature on MET exon 14 skipping (https://pubmed.ncbi.nlm.nih.gov/) we observed that the identification of this skipping event is usually done using DNA-based amplicon-mediated target enrichment [15] or RNA-based next-generation sequencing target enrichment [16], where the RNA based method provides a higher detection rate of exon 14 skipping [16]. We never found any article using deep learning or machine learning methods for the detection of MET exon 14 skipping, which is of particular interest to screen large cohorts of specimens.  Notably, we found two articles [17] [18] predicting exon skipping events using RNAseq data. In Zhang’s paper [17] a convolutional neural network (CNN) is used to classify splice junctions derived from primary RNA-seq data. Instead in Du’s paper [18] a Rotation Forest algorithm is used to predict ES events integrating RNA-Seq data and genome sequence information. Moreover, CNN was implemented in SpliceRover [19] a generalist tool for splice site prediction. Similarly, in [20] a general tool namely SpliceAI was proposed to predict splicing from a pre-mRNA sequence using CNN.

To the best of our knowledge, we did not find any tool designed specifically to detect an exon skipping event as MET exon 14 skipping. In this manuscript, we challenged different neural network architectures to provide sensitive and rapid detection of MET exon 14 skipping events using RNAseq data. Here, we discuss the results obtained comparing a deep learning algorithm, such as a convolutional neural network (CNN), with respect to a conventional neural network (NN) and a sparsely connected autoencoder (SCA). With respect to the above-mentioned models, predicting ES events, as well as other generalist CNNs [21, 22], which are designed using nucleotide sequence information for their prediction, our models are designed to handle expression data in the form of kmer counts or coverage.”

Q2: A main point/contribution of this paper is the use of deep learning. However, the authors did not mention it a lot in the methodology as well as in the results. More analyses on deep learning architecture should be performed.

Answer: We added to the manuscript the implementation of a convolutional neural network and we compared the performance of the CNN with respect to NN discussing the different results obtainable using different training and test sets.

In introduction we rewrote the following sentence:

“In this manuscript, we describe the development and validation of a neural network (NN) specifically devoted to the effective and rapid detection of MET exon 14 skipping events using RNAseq data.”

in

“In this manuscript, we investigated different neural network architectures to provide sensitive and rapid detection of MET exon 14 skipping events using RNAseq data. Standard  Neural Network (NN), Convolutional Neural Network (CNN), and a Sparsely Connected Autoencoder (SCA) were thus compared in detail on different dataset..

In results we added:

“ 2.2. Convolutional neural network (CNN) for the detection of METD14.

To detect MET exon 14 skipping events, we constructed a CNN made by a 1D convolutional layer, 1D Max pooling layer, a flat fully connected dense layer with 50 nodes and an output layer with one node, Figure 1C. The CNN was challenged with the same training and test set used for the flat neural network. In this implementation, the convolutional layer included 10 kernels, for more information see Material and Method section. In Figure 5, the METD14 detection ability of CNN on the basis of different representation of the MET expression data are reported. The results are organized, Figure 5, on the basis of the type of input data, i.e. whole MET exons kmer counts (Figure 5A), MET exons 13-15 kmer count frequency (Figure 5B) and MET exons 13-15 coverage frequency (Figure 5C). The best ratio between true positive and false positive is shown for all kernels using test samples characterized by 5000 reads coverage. As also seen for NN, Figure 3, the specificity progressively decreases when the coverage is reduced.

2.2.1 Convolutional neural network validation on bronchus and lung samples.

We validated the CNN model using the kernel 100, which is one of the best performing kernels independently by the coverage of the test set (Figure 5). The validation was done on the 690 TCGA bronchus and lung sample manually inspected for the presence of METD14. We tested on this tumor set the CNN trained with k-mer counts, which predicted 10 samples as METD14, but only one was a real exon skipping events (sensitivity 5.88%, specificity 97.6%, supplementary table 6S). The best results were obtained using the CNN trained with exons 13¸15 MET k-mer counts frequency. All of the 16 samples predicted as METD14 belong to the 17 true METD14 present in the data set (sensitivity 94.11%, specificity 100%, supplementary table 6S). Finally, the CNN trained using only the coverage frequency for MET exons 13¸15, predicted 8 skipping events and all of them belong to the true skipping events (sensitivity 47.05%, specificity 100%, supplementary Table 6S). Since we observed that NN was detecting some false positives in cervix tumor tissues, Figure 4, we evaluated if CNN was more specific than NN. CNN trained with exons 13¸15 MET k-mer counts frequency detects the same false positives detected by NN, Figure 4.”

In Material and Methods we added:

“4.5. Model coding and hyperparameter selection for CNN

We constructed a CNN made of one Convolutional 1D layer, characterized by 64 filters and the following kernel sizes: 2, 5, 7, 10, 15, 50, 75, 100, 150, 200. One MaxPooling 1D layer with pool size of 2, for dimensionality reduction. One dense layer with activation RELU and 50 nodes and one dense layer with activation sigmoid and 1 node. We implemented the models in python (version 3.7) using TensorFlow package (version 2.0.0), Keras(version 2.3.1), pandas (version 0.25.3), numpy (version 1.17.4), matplotlib(version 3.1.2), sklearn (version 0.22), scipy (version 1.3.3). Optimization was done using Adam (Adaptive moment estimation) using the following parameters lr=0.01, beta_1=0.9, beta_2=0.999, epsilon=1e−08,decay=0.0, loss=‘mean_squared_error’”

Q3: The authors should compare the predictive performance of their models with previous studies on the same problem/dataset. Also, some baseline comparisons should be conducted to convince that the authors' choice is the optimal one.

Answer: We added the following sentences in Discussion:

“It is notable that, digging into the published literature, all exon skipping tools found use nucleotide sequence analysis to infer skipping events, and they are only able to predict skipping events in a generalist way. Since, we could not find any tool providing the detection of a unique skipping event in a gene over a large cohort of specimens, we designed specific neural networks for the identification of MET exon 14 skipping, using transcript expression information.”

Q4: How did the author deal with hyperparameter optimization of the models? It should be described clearly.

Answer: We inserted the following phrase in the Material and Methods associated with CNN and NN description: “Hyperparameter optimisation was done using Talos (https://github.com/autonomio/talos), which is an automated tool to define the optimal combination of the hyperparameters. Specifically, Talos takes as input the hyperparameter space to be investigated. Then, Talos performs all possible combinations and selects the optimal configuration of the hyperparameters.”

In discussion we added the following phrase: “Although we performed an automated optimization of the hyperparameters, the prediction efficacy of our CNN and NN comes from the special attention we put on defining the optimal representation of the data for each architecture, i.e kmer counts for CNN and coverage from NN.”

Q5: How did the authors deal with imbalance problem? Sometimes I have seen that the authors reached a very imbalance result (supplementary table S2).

Answer: We extended the discussion section: “… Skewed datasets are not uncommon and the MET exon 14 skipping detection is a typical example. Although skewed datasets are tough to handle, our models, i.e. CNN and NN, seem to handle this issue efficiently, since sensitivity greater than 94% and specificity greater than 99% are reached on an extremely skewed data set such as TCGA bronchus and lung 690 samples with only 17 MET exon skipping events (2.46%). Notably, the high sensitivity is obtained by CNN with a training based on kmer counts spanning among MET exon 13 and exon 15. Instead, in the case of the NN the optimal sensitivity was obtained with a training based on coverage data encompassing the region among MET exon 13 and exon 15.

Our analysis, using both CNN and NN, on 2605 TCGA tumors (13 primary sites, Table 2) highlights that MET exon 14 skipping is a peculiar event of lung specimens. Then, mainly in uterine cancers, we detected a set of MET exon 14 skipping false positives, sharing a common feature: an unexpected peak of coverage in the MET intron 2. This observation brought us to speculate that we were observing a transcriptional signature for a LINE1-MET fusion event [30]. This hypothesis has been supported by the identification of MET paired-end reads, having one read mapping on MET and the other on LINE1 sequence. Notably, transcription of the LINE1-MET fusion was observed in advanced stages of cancer [30, 39], but very little is still known about the effect of the LINE1-MET chimera in cancer.

At the present time, we cannot manage to eliminate LINE1-MET false positives, mainly because we do not have enough data to train a model to detect LINE1-MET fusion, to be implemented in parallel with the MET exon 14 skipping models. However, we are generating large RNAseq data from MCF7, a breast cancer cell line harboring LINE1-MET fusion [30], to build a specific neural network model to be integrated with our MET exon 14 skipping models, to improve its specificity.”

Q6 The tool cannot be accessed. Please re-check the link.

Answer: We have modified the MetObservatory github, providing a better explanation of the models usage.

Q7: Deep learning, or DNN, has been used in previous studies i.e., PMID: 33260643 and PMID: 31750297. Therefore, the authors are suggested to refer to more works in this description.

Answer: We thank the reviewer for the kind suggestions. The two suggested papers were very useful to better focus our article. Please see the changes we have made in the introduction section as an answer to your Q1.

Q8: Quality of figures should be improved significantly.

Answer: Figures were revised improving their quality as required

Q9: More validation data should be used.

Answer: As a validation set, we selected the 1310 tumor samples available at TCGA from the bronchus and lung primary sites. The selection of these specific tumor primary sites is due to the observations that MET exon 14 skipping is mainly present in lung cancer and it is expected to be found with a frequency of 2.3-2.6% in such dataset. Actually, we detected 17 MET exon 14 skipping in 690 tumors (2.43%). The validation set was restricted to 690 samples because only those, out of the 1310 bronchus and lung tumors available TCGA samples, were providing coverage of at least 5000 reads at the MET locus.

Furthermore, we analyzed 2605 TCGA tumors (13 primary sites). These 2605 tumors are the subset of 8638 TCGA tumors, which provide a coverage of at least 5000 reads on the MET locus.

I am very sorry but we cannot further extend our analysis on other data, because we do not have access to any data other than TCGA.

Reviewer 2 Report

Authors demonstrated the neural networks that can be a tool for quick classification of pathological transcription events. 

  1.  genetic variants, deep learning could be included in Keywords.
  2. Abbreviations should be explained at their first appearances. For example, LIN1 in line26, ALK, APC, BAP1, BRCA1, etc. in line 45 and 47. 
  3. Figure 5 should be enlarged.
  4. Please indicate the source of the cell lines in line 205-207.
  5. Why authors chose the cell lines to use in this experiments? In introduction section authors mentioned the MET and NSCLC relation. What about the gastric cancer cell line? Please explain this in discussion section.
  6. Discussion section is too short. Please explain the meaning of the experiments in depth. Also, compare other previous studies to current study.

Author Response

We would like to thank the reviewers for their valuable comments and useful suggestions that helped us to substantially improve the quality of this paper.

Reviewer 2

Authors demonstrated the neural networks that can be a tool for quick classification of pathological transcription events. 

Q1: genetic variants, deep learning could be included in Keywords.

Answer: As suggested by reviewer we added them to Keywords

Q2: Abbreviations should be explained at their first appearances. For example, LIN1 in line26, ALK, APC, BAP1, BRCA1, etc. in line 45 and 47.

Answer: We added the full name of each gene

Q3: Figure 5 should be enlarged.

Answer: Figure was revised and enlarged as required.

Q4: Please indicate the source of the cell lines in line 205-207.

Answer: We added in Material and Method:A549 (lung adenocarcinoma); NCI‐H596 (lung adenocarcinoma); Hs746T (gastric adenocarcinoma) cell lines were purchased from ATCC (Rockville, MD); EBC‐1 (non‐small cell lung cancer) were acquired from HSRRB cell bank (Osaka, Japan). All cells were kept in culture for less than 4 weeks and used between passage 2 and 10. Cells were grown in recommended media (Sigma Aldrich, St. Louis, MO) supplemented with 50 units/mL penicillin (Sigma Aldrich, St. Louis, MO), 50 mg/mL streptomycin (Sigma Aldrich, St. Louis, MO), 2 mM glutamine (Sigma Aldrich, St. Louis, MO) and 10% Foetal Bovine Serum (Lonza Sales Ltd, Basel, Switzerland) as indicated. Cells were maintained at 37 °C in a 5% CO2 atmosphere. “

Q5: Why authors chose the cell lines to use in this experiments? In introduction section authors mentioned the MET and NSCLC relation. What about the gastric cancer cell line? Please explain this in discussion section.

Answer: We added the following sentences in the discussion: “The NN and the CNN training was done using the RNAseq data of a lung cancer cell line expressing amplified form of the wild type MET (WT, EBC‐1), and a gastric cancer cell line expressing exon 14 skipped MET (HS746T).  HS746T cell line was selected because, to the best of our knowledge, it is the only cell line displaying amplification of MET exon 14 skipping isoform. MET gene amplification has been observed in about 2-5% of gastroesophageal cancers and represents an oncogenic driver and therapeutic target [36, 37]. MET exon 14 skipping was initially described in NSCLCs (caused by a mutation in the splice donor site in intron 14 and afterwards reported in a variety of tumors, including gastrointestinal cancers, suggesting it as a potential mechanism leading to MET activation [38]. Therefore, HS746T, together with EBC‐1, was an invaluable instrument to provide a large amount of data for the NN/CNN training. Validation was done instead using RNAseq data from lung cancer cell lines expressing at physiological level MET (A549 expressing WT MET and NCI‐H596 expressing exon 14 skipped MET).”

Q6: Discussion section is too short. Please explain the meaning of the experiments in depth. Also, compare other previous studies to current study.

Answer:  Discussion section was extended to better explain the experiments and the results:

“We used MET exon 14 skipping as a case study for the detection of genetic variants in cancer driver genes by deep learning. In recent years, a lot of evidence indicates that MET inhibitors have a good anti-tumor effect in patients with MET exon 14 skipping mutation, suggesting that MET exon 14 skipping may be a new target for NSCLC patients [35]. Thus, the availability of effective tools for the detection of MET exon 14 skipping are needed for a fast identification of patients suitable for MET targeted therapy.

It is notable that, digging into the published literature, all the found exon skipping tools use nucleotide sequence analysis to infer skipping events, and they are only able to predict skipping events in a generalist way. Since, we could not find any tool providing the detection of a unique skipping event in a gene over a large cohort of specimens, we designed specific neural networks for the identification of MET exon 14 skipping, using transcript expression information.

We designed a conventional neural network (NN) made of four fully connected hidden layers and a convolutional neural network (CNN) made of one 1D convolutional layer, one 1D max pooling layer and a fully connected dense layer. Although we performed an automated optimization of the hyperparameters, the prediction efficacy of our CNN and NN comes from the special attention we put on defining the optimal representation of the data for each architecture, i.e kmer counts for CNN and coverage from NN.

The NN and the CNN training was done using the RNAseq data of a lung cancer cell line expressing amplified form of the wild type (WT, EBC‐1), and a gastric cancer cell line expressing exon 14 skipped MET (HS746T).  HS746T cell line was selected because, to the best of our knowledge, it is the only cell line displaying amplification of MET exon 14 skipping isoform. MET gene amplification has been observed in about 2-5% of gastroesophageal cancers and represents an oncogenic driver and therapeutic target [36, 37]. MET exon 14 skipping was initially described in NSCLCs (caused by a mutation in the splice donor site in intron 14 and afterwards reported in a variety of tumors (including gastrointestinal cancers) suggesting it as a potential mechanism leading to MET activation [38]. Therefore, HS746T, together with EBC‐1, was an unevaluable tool to provide a large amount of data for the NN/CNN training. Validation was done instead using RNAseq data from lung cancer cell lines expressing at physiological level MET (A549 expressing WT MET and NCI‐H596 expressing exon 14 skipped MET).

Since we could not compare our models with respect to pre-existing methods for MET exon 14 skipping, we manually curated a set of TCGA data, to provide an objective evaluation of the performance of our tool. Specifically, we manually curated a cohort of WT and exon 14 skipped samples made of the 690 RNAseq samples belonging to the TCGA (https://www.cancer.gov/tcga) bronchus and lung collection (1310 samples) showing a MET coverage of at least 5000 reads. Given the manual curation of this dataset, i.e. each single sample was inspected on IGV browser for the presence of MET exon 14 skipping, it represents a robust instrument to quantify the predictive performance of our neural network models.

Skewed datasets are not uncommon and the MET exon 14 skipping detection is a typical example. Although skewed datasets are tough to handle, our models, i.e. CNN and NN, seems to handle this issue efficiently reaching sensitivity greater than 94% and specificity greater than 99% on an extremely skewed data set such as the TCGA bronchus and lung 690 samples with only 17 MET exon skipping events (2.46%). Notably, this high sensitivity is obtained by CNN with a training based on kmer counts spanning among MET exon 13 and exon 15. Instead, in the case of the NN the optimal sensitivity was obtained with a training based on coverage data encompassing the region among MET exon 13 and exon 15.

Our analysis, using both CNN and NN, on 2605 TCGA tumor (13 primary sites, Table 2) highlights that MET exon 14 skipping is a peculiar event of lung specimens. Then, mainly in uterine cancers, we detected a set of MET exon 14 skipping false positives, sharing a common feature: an unexpected peak of coverage in the MET intron 2. This observation brought us to speculate that we were observing a transcriptional signature for a LINE1-MET fusion event [30]. This hypothesis has been supported by the identification of MET paired-end reads, having one read mapping on MET and the other on LINE1 sequence. Notably, transcription of the LINE1-MET fusion was observed in advanced stages of cancer [30, 39], but very little is still known about the effect of the LINE1-MET chimera in cancer.

At the present time, we cannot manage to eliminate LINE1-MET false positives, mainly because we do not have enough data to train a model to detect LINE1-MET fusion, to be implemented in parallel with the MET exon 14 skipping models. However, we are generating large RNAseq data from MCF7, a breast cancer cell line harboring LINE1-MET fusion [30], to build a specific CNN to be integrated with our MET exon 14 skipping models, to refine its specificity.

Having identified more than one artifactual event in MET, we investigated the possibility to discover those anomalous events by the integration of a particular type of deep learning tool, sparsely connected autoencoders [32], with clustering techniques used in multicolor cytometry. Although the actual implementation of the SCA tool could be further improved in terms of its precision and sensitivity, currently we were able to detect from TCGA specimens a set of tumors sharing the putative LINE1-MET fusion.

Taken together our results indicate that neural networks can be an effective tool to provide a quick classification of pathological transcription events. However, from the discovery point of view there is still some work to be done to obtain an effective discovery tool using sparsely connected autoencoders.”

Round 2

Reviewer 1 Report

My previous comments have been addressed well.

Reviewer 2 Report

Authors revised the manuscript according to my suggestion.

This study is good to be published.